# Metabolic Reprogramming in Melanoma: An Epigenetic Point of View

**DOI:** 10.3390/ph18060853

**Published:** 2025-06-06

**Authors:** Stefano Giuliani, Celeste Accetta, Simona di Martino, Claudia De Vitis, Elena Messina, Edoardo Pescarmona, Maurizio Fanciulli, Gennaro Ciliberto, Rita Mancini, Italia Falcone

**Affiliations:** 1SAFU, Department of Research, Advanced Diagnostics and Technological Innovation, IRCCS-Regina Elena National Cancer Institute, 00144 Rome, Italy; maurizio.fanciulli@ifo.it (M.F.); italia.falcone@ifo.it (I.F.); 2Department of Pathology, IRCCS-Regina Elena National Cancer Institute, 00144 Rome, Italy; celeste.accetta@ifo.it (C.A.); edoardo.pescarmona@ifo.it (E.P.); 3Department of Clinical and Molecular Medicine, Sant’Andrea Hospital, “Sapienza” University of Rome, 00161 Rome, Italy; claudia.devitis@uniroma1.it (C.D.V.); elena.messina@uniroma1.it (E.M.); rita.mancini@uniroma1.it (R.M.); 4Takis s.r.l., Via di Castel Romano 100, 00128 Rome, Italy; gennaro54.ciliberto@gmail.com

**Keywords:** melanoma, metabolism, epigenetics, drug resistance, target therapy, immunotherapy

## Abstract

Metabolic reprogramming and epigenetic alterations are fundamental hallmarks of cancer cells, contributing to adaptation, progression, and resistance. In melanoma, high metabolic-epigenetic plasticity enables the rapid modulation of cell states in response to environmental and therapeutic pressures. Recent studies have highlighted a bidirectional crosstalk between cellular metabolism and epigenetic regulation. Epigenetic modifications influence the transcriptional control of metabolic genes, thereby shaping metabolic phenotypes. Conversely, specific metabolites are essential cofactors or substrates for epigenetic enzymes, directly modulating the epigenome. Understanding the intricate mechanisms of this interaction offers opportunities for the development of innovative tumor management that combines epigenetic, metabolic, and therapy interventions. In this review, we summarize the latest evidence on the role of the metabolism–epigenetics axis in melanoma and discuss its potential clinical implications, aiming to provide a comprehensive overview of metabolic/epigenetic interconnections.

## 1. Introduction

Melanoma represents one of the most aggressive and lethal forms of cancer, characterized by high genetic and metabolic heterogeneity. Accordingly, metabolic reprogramming is one of the hallmarks of melanoma and has emerged as a key element in tumor development, disease recurrence, metastasis, and drug resistance [1]. The metabolic plasticity of melanoma allows tumor cells to rapidly and efficiently adapt to external stimuli, including therapeutic pressures. This phenomenon reflects the tumor complexity composed of a highly heterogeneous population, allowing the coexistence of different metabolic and mutational states and drastically increasing the difficulty of developing therapeutic strategies [2]. In the current therapeutic landscape, immune checkpoint inhibitors (ICIs) targeting PD-1 and CTLA-4, as well as targeted therapies against MAPK pathway components (e.g., BRAF and MEK inhibitors), have become standard-of-care treatments in melanoma. Nonetheless, recurrence remains a significant clinical challenge, with relapse rates ranging from 20 to 40% in initially responsive patients, due to the emergence of resistance mechanisms and tumor plasticity [3,4].There is growing evidence that non-genetic events underline cell state transitions and state switching, allowing cancer cells to develop resistance, and altered epigenetic mechanisms have been linked to melanoma development and progression [5]. In light of this, we hypothesize that the bi-directional interface between metabolism and epigenetics represents a crucial node underlying melanoma’s high adaptive capacity and resistance to therapies. Therefore, this review aims to address the therapeutic challenges outlined above by examining in detail the interplay between metabolism and epigenetics in melanoma. First, we will analyze the main metabolic reprogramming pathways in melanoma, focusing on the molecular mechanisms that enable its plasticity. Next, we will examine the epigenetic modifications that orchestrate cell state transitions and alter the expression of metabolic regulators. Finally, we will discuss how metabolites serve as substrates and modulators of epigenetic enzymes, representing potential therapeutic targets for integrated strategies that simultaneously target oncogenic signals, metabolic networks, and the epigenome. The goal is to provide a comprehensive overview of metabolic-epigenetic interconnections in melanoma.

## 2. Melanoma Metabolic Reprogramming

In differentiated cells, such as melanocytes, energy production is mainly driven by mitochondrial oxidative phosphorylation (OXPHOS). This complex and efficient process not only maximizes ATP generation but also maintains redox balance and supports cellular functions, such as pigment synthesis and cellular homeostasis [6]. This metabolic pathway ensures a control and oxidative stress management of “normal melanocytes”, and provides for their bioenergetic requirements [7]. Energy metabolism reprogramming has long been considered one of the hallmarks of cancer [8,9]. Several studies have shown that cancer cells prefer glycolytic metabolism rather than exploiting the full capacity of their mitochondria, even in the presence of oxygen [10,11,12,13]. This phenomenon was described historically as early as 1924 as the Warburg effect, named after its discoverer [14]. Despite the lower ATP efficiency of glycolysis compared to OXPHOS, this metabolic change provides significant benefits to tumor cells, as follows: (i) rapid ATP production to sustain high proliferative rates; (ii) generation of biosynthetic precursors for macromolecule synthesis; (iii) acidification of the tumor microenvironment (TME) due to lactate accumulation; (iv) therapeutic evasion; and (v) metastatic propagation [15,16]. Melanoma, a highly aggressive and metabolically heterogeneous neoplasm, undergoes profound metabolic reprogramming to sustain rapid proliferation, invasion, and therapeutic resistance. Among the key metabolic pathways reprogrammed in melanoma, glycolysis, OXPHOS, and lipid metabolism play crucial roles in sustaining tumor growth, dissemination, and adaptation to microenvironmental stresses [17].

### 2.1. Melanoma Metabolic Plasticity: Glycolysis vs. OXPHOS

Cutaneous melanoma is one of the most aggressive and therapy-resistant cancers and exhibits remarkable metabolic plasticity. In melanoma, the primary driver mutations occur in v-raf murine sarcoma viral oncogene homolog B (BRAF (40–50%) and neuroblastoma RAS viral oncogene homolog (NRAS) (15–30%) [18,19], which are key components of the mitogen-activated protein kinase (MAPK) cascade and are associated with increased glucose uptake and fuel aerobic glycolysis [20,21]. Specifically, MAPK activation is directly linked to the induction of master transcriptional regulators of glycolysis, such as hypoxia-inducible factor 1α (HIF-1α) and c-Myc. These regulators coordinate the expression of several key enzymes in glucose metabolism, thereby shifting cellular energy production from OXPHOS to glycolysis, even under normoxic conditions [22]. Mechanistically, the interaction of HIF-1α with HIF-1β leads to the transcriptional activation of genes, such as lactate dehydrogenase (LDH), aldolase, and enolase 1 (ENO1), increasing glycolytic flux. At the same time, HIF-1α inhibits pyruvate dehydrogenase (PDH) by activating pyruvate dehydrogenase kinase (PDK), leading to a reduced conversion of pyruvate to acetyl-CoA and promoting lactic fermentation, thereby increasing lactate production [23,24,25,26]. In maintaining aerobic glycolysis, c-Myc plays a key role by directly activating the transcription of almost all glycolytic genes and acting on transmembrane transporters of glucose to increase its uptake. It also contributes to the increased acidification of the TME by modulating the monocarboxylate transporters (MCT1 and MCT2), increasing toxic lactate levels [27,28]. Another key protein in glycolytic metabolism is pyruvate kinase M2 (PKM2), a regulatory isoenzyme that catalyzes the last step of glycolysis. PKM2 is up-regulated and allows cancer cells to direct metabolites to biosynthesis [29]. Furthemore, PKM2 not only promotes proliferation, but is also involved in non-canonical epigenetic functions, including histones phosphorylation (e.g., H3-T11), influencing gene expression in response to metabolic stimuli [30].

In melanoma, the pharmacological inhibition of PKM2 by Benserazide has been shown to inhibit tumor growth and enhance the efficacy of BRAFi, suggesting possible combined use in target therapy [31].

The increase in intracellular glucose is exploited by the tumor cell through the pentose phosphate pathway (PPP). Glucose metabolism generates nicotinamide adenine dinucleotide phosphate (NADPH), essential for catalyzing the glutathione oxidation reaction (GSSG) to glutathione (GSH), providing to the cell with a defense against oxidizing agents [32]. Indeed, it has been observed that loss of the glucose 6-phosphate dehydrogenase (G6PD) function, a key enzyme in PPP, increases oxidative stress and glutaminolysis in metastatic melanoma cells [33]. Moreover, BRAF suppresses the microphthalmia-associated transcription factor (MITF) and peroxisome proliferator-activated receptor-γ coactivator 1α (PGC1α), further contributing to the downregulation of OXPHOS [34].

While glycolysis is often dominant in melanoma, OXPHOS remains an important metabolic strategy, particularly in therapy-resistant and invasive melanoma subpopulations. Mitochondrial respiration, driven by the tricarboxylic acid (TCA) cycle and the electron transport chain (ETC), provides a more efficient ATP production and allows melanoma cells to utilize alternative carbon sources such as fatty acids and glutamine, offering a survival advantage under nutrient-deprived conditions [35]. In this context, MITF plays a key role in the regulation of mitochondria energy production by interacting directly with the mitochondrial master regulator PGC1α. It is now known that the MITF/PGC1α axis in melanoma represents a key node in the metabolic switch of the tumor cell, allowing a fast adaptive response to external stimuli [36]. Ho et al. demonstrated that melanoma exploits both glycolysis and OXPHOS through three distinct cell populations. This metabolic symbiosis highlights melanoma’s adaptability and its ability to grow in diverse microenvironments [37]. However, contrary to conventional ideas, melanoma cannot simply diversify into “glycolytic” or “OXPHOS” cells; indeed, several studies have shown the possibility of a hybrid phenotype, giving cancer cells strong advantages [38].

Metabolic plasticity in melanoma not only supports its growth and adaptation to various microenvironments but also plays a critical role in therapy resistance. Although targeted therapies initially show high efficacy in BRAF-mutant melanoma, they frequently develop resistance due to metabolic rewiring [39,40]. Smith and collaborates observed OXPHOS metabolism activation in Braf-mutated cells following BRAF inhibitor (BRAFi) treatment and linking this phenomenon to a translational metabolic reprogramming event [41]. Similarly, melanoma cells can exploit metabolic adaptation to evade immunotherapy. Immune checkpoint inhibitors (ICIs) rely on an active immune microenvironment. But the metabolic reprogramming, such as increased lactate production, creates an immunosuppressive niche, reducing T cell function and dampening the immune response [42,43]. Furthermore, the accumulation of metabolites like lactate and adenosine not only inhibits T cell proliferation and cytotoxicity but also promotes macrophage polarization toward an M2-like immunosuppressive phenotype, further contributing to resistance to ICIs [44]. These processes highlight how metabolic flexibility represents a major hurdle in melanoma treatment, necessitating novel therapeutic strategies that target both signaling and metabolic vulnerabilities.

### 2.2. Lipid Metabolism in Melanoma

Lipid metabolism plays a central role in tumorigenesis and progression in different tumor contexts [45]. Lipid metabolism underlies cellular homeostasis and is regulated by a fine balance between lipid uptake, synthesis, and hydrolysis. In the melanoma context, the perturbation of this balance promotes events such as cell proliferation, survival, invasion, and metastasis. Generally, this balance is maintained by the de novo synthesis of fatty acids (FAs) and by fatty acids (FAs) β-oxidation [46,47]. In lipogenesis, the main substrate is acetyl-CoA generated through the conversion of glucose-derived citrate by the action of the ATP-citrate lyase (ACLY) [48,49]. ACLY represents a key enzyme in the link between glycolysis and lipogenesis. Several studies showed that in melanoma, the deregulation of this enzyme underlies events such growth [50], progression [51], and therapy evasion [50,52]. Another key mechanism in FAs’ generation lies in the deregulation of lipid homeostasis between Saturated Fatty Acids (SFAs), Monounsaturated Fatty Acids (MUFAs), and Polyunsaturated Fatty Acids (PUFAs). In this context, a key enzyme is Stearoyl-CoA desaturase (SCD1), which catalyzes the desaturation of SFAs into MUFAs, unbalancing their rate. In different tumor contexts, it has been observed that increasing SCD1 prevents the accumulation of SFAs by preventing endoplasmic reticulum stress phenomena and apoptosis [45,53]. In melanoma, MITF directly regulates SCD1 expression and MITF^higt^ cells are more susceptible to SCD1 treatment by blocking their proliferation [54]. On the other hand, several studies that have examined SCD1 activity have identified a correlation between lipid metabolism and cancer stem cells (CSCs) [55,56]. On this basis, the pharmacological inhibition of SCD1, in a BRAF-mutated melanoma context, showed a restored response to target therapy in 3D cultures and could be considered a new potential biomarker in treatments of melanoma CSCs [57]. In addition to lipogenesis, fatty acid oxidation mediated by the FAO enzyme in mitochondria also plays a key role in melanoma, especially in the metabolic switch and drug resistance. Aloia and collaborates demonstrated how targeted therapies directly inhibit glycolysis, inducing the tumor cell to make a metabolic switch to an OXPHOS phenotype. This involves an activation of FAO and fatty acid transporter CD36, thereby using FAs as the main source of energy [58]. A recent study showed that the combination between ranolazines (FAO inhibitor) and BRAFi delays the onset of acquired resistance and appears to sensitize to immunotherapy by promoting an immunogenic phenotype in melanoma [43].

### 2.3. Amino Acid Metabolism in Melanoma

Amino acid metabolism—including glutamine, arginine, and serine pathways—plays an important role in melanoma progression and drug resistance [59]. Glutamine, the most abundant non-essential amino acid, serves as a major anaplerotic substrate by converting into glutamate and further into α-ketoglutarate (α-KG) to replenish the TCA cycle [60]. In melanoma, the expression of glutamine transporters, particularly ASCT2 (SLC1A5), is significantly elevated, facilitating enhanced glutamine uptake. ASCT2 inhibition has been shown to reduce glutamine, leading to suppressed mammalian target of rapamycin complex 1 (mTORC1) signaling and decreased melanoma cell proliferation [61,62].

Arginine metabolism is equally pivotal because increased arginine turnover results in the production of nitric oxide (NO) via nitric oxide synthases (NOSs), promoting angiogenesis, invasion, and immune escape [63,64]. Moreover, the upregulation of arginase 1 (Arg1) in tumor-associated macrophages leads to polyamine synthesis, stimulating melanoma-initiating cell growth and contributing to an immunosuppressive microenvironment [65].

Serine metabolism, through the de novo serine synthesis pathway (SSP), provides critical building blocks for protein synthesis and one-carbon metabolism [66,67]. Key enzymes such as D-3-Phosphoglycerate dehydrogenase (PHGDH), phosphoserine aminotransferase 1 (PSAT1), and phosphoserine phosphatase (PSPH) are upregulated in melanoma, enhancing nucleotide synthesis, methylation reactions, and antioxidant defense via glutathione production [68,69]. Collectively, the interplay between these amino acid pathways ensures melanoma cells meet their biosynthetic and redox demands, while also promoting immune evasion and therapeutic resistance.

## 3. Epigenetic Regulation of Metabolic Plasticity

In recent years, cancer research showed that epigenetic mechanisms promote metabolic state transitions, influencing cell growth, adaptation, and resistance to therapies [70]. DNA methylation, histone modifications, and regulation by noncoding RNAs represent the most important epigenetic modifications. These events allow the cancer cell to change its gene set in a heritable and reversible manner without altering the DNA sequence [71,72,73]

### 3.1. DNA Methylation

DNA methylation involves the addition of a methyl group (-CH_3_) to a DNA nitrogenous base, usually to cytosine in a CpG (cytosine–guanine dinucleotide) context. CpG island-rich regions are often found close to gene promoters, leading to the epigenetic regulation of their transcription. This process is catalyzed by two enzyme families: DNA methyltransferases 1–3 (DNMT 1–3) responsible for methylation and its maintenance, and translocation (TET) methylcytosine dioxygenases involved in the reverse process of demethylation [74,75]. In the cancer context, the deregulation of this process is often observed, mainly affecting onco-suppressor genes, thus promoting tumor growth and survival [76]. In tumors, DNA methylation has been shown to influence tumor metabolism both directly, by methylating key metabolic genes, and indirectly, by targeting tumor suppressor genes involved in signaling pathways related to tumor metabolism [77]. Aberrant patterns of DNA hyper-methylation are often identified in melanoma [78,79,80] (Figure 1A).

For example, the hypermethylation of the phosphatase and tensin homolog (PTEN) gene is directly related to lower overall survival (OS) in melanoma patients, which also correlates with increased tumor thickness (Breslow) and increased frequency of ulceration [81,82]. PTEN is a negative regulator of the PI3K/AKT pathway [83], and its silencing or loss of function promotes cancer cell growth and survival, enhancing glycolysis [78,84]. In addition, in vivo studies showed that DNMT1 promotes tumorigenesis in melanoma cells by the direct activation of the PI3K/AKT/mTOR pathway. This occurs through the repression of the interaction between two inhibitory molecules, HSPB8 and BAG3 [85].

Another gene that often exhibits epigenetic regulation by methylation is the master regulator MITF. From analysis of the genome-wide DNA methylation profiles of 50 patients with stage IV melanoma compared with normal melanocytes, keratinocytes, and dermal fibroblasts, Lauss et al. observed the hypermethylation of MITF and its co-regulated differentiation pathway genes, with decreased gene expression levels. Indeed, in MITF-hypermethylated melanoma cell lines, a demethylation treatment induces a reactivation of the MITF pathway [86]. Melanoma presents itself as a highly heterogeneous tumor characterized by a very high cellular plasticity. This peculiarity allows it to switch from a proliferative to an invasive mesenchymal state and vice versa using MITF as a “switch.” Low levels of MITF are generally linked to an invasive and resistant metastatic phenotype; conversely, high levels of MITF are linked to a proliferative phenotype [35,87]. Cells with different metabolic, epigenetic, and transcriptional arrangements can coexist, making this switching phenomenon extremely complex to overcome.

DNA hypomethylation also plays a role in the activation of genes involved in tumor metabolism, such as Transketolase-like 1 (TKTL1) (Figure 1A). TKTL1 links the PPP with the glycolytic pathway, genetically silenced by promoter methylation in normal somatic tissues [32,88]. In melanoma, however, promoter hypomethylation results in the aberrant expression of TKTL1, promoting a glycolytic phenotype. The elevated expression of TKTL1 increases glucose use and lactate production; consequently, melanoma cells with high TKTL1 expression show increased proliferation and metastatic potential, demonstrating a connection between aberrant DNA methylation patterns and altered metabolic pathways [89].

In summary, alterations in DNA methylation significantly influence melanoma progression by modifying key genes associated with tumor metabolism. These epigenetic changes facilitate metabolic adaptation that supports tumor growth, invasiveness, and resistance to therapies.

### 3.2. Histone Modification

Histones, the core structural proteins of chromatin, have flexible N-terminal tails rich in lysine and arginine residues that are subject to diverse covalent modifications. Among these modifications, histone acetylation and methylation have been extensively studied due to their pivotal roles in gene expression control and their involvement in cancer biology. These modifications act as dynamic epigenetic marks, orchestrating complex transcriptional responses involved in tumor progression, tumor resistance, and metabolic adaptation [90,91]. Histone acetylation, mediated by the antagonistic activities of histone acetyltransferases (HATs) and histone deacetylases (HDACs), generally leads to chromatin relaxation and active transcription [92]. In melanoma, the aberrant regulation of HATs and HDACs contributes to metabolic plasticity, therapy resistance, and the dysregulation of key tumor suppressors and oncogenes [93] (Figure 1B). HDAC8 promotes melanoma growth and migration by acting HIF-1α. Specifically, HDAC8 increases the stability of HIF-1α via histone deacetylation, thereby increasing its ability to activate target genes associated with tumor progression by boosting glycolysis and metastasis formation [94]. Furthermore, a pharmacological approach based on compounds that contemporary target HDAC and the σ1 receptor has recently been proposed. These compounds have shown selective efficacy against cutaneous and uveal melanoma cell lines, suggesting how targeted treatment on HDAC may have significant clinical benefits, especially when combined with anti-angiogenic strategies via the modulation of the σ1 receptor [95]. Sirtuins (SIRTs), NAD-dependent class III deacetylases, have emerged as regulators of epigenetic modifications in melanoma, acting on several cellular processes like oxidative stress, response to UV damage, aging, and metabolism (Figure 1B). SIRT1, often overexpressed in melanoma, influences cell proliferation and survival by regulating the acetylation of key targets such as p53 and p21, affecting cell cycle and metabolism. The pharmacological inhibition of SIRT1 reduces melanoma cell growth, suggesting its therapeutic potential [85]. SIRT5 modulates histone acetylation and methylation, maintaining the expression of critical genes such as MITF and c-Myc, thus promoting melanoma cell survival and representing a genotype-independent therapeutic vulnerability [86]. Furthermore, SIRT6, significantly upregulated in melanoma, is implicated in tumor progression through the regulation of autophagy, cellular senescence, and metabolic adaptation, underscoring its role as a potential oncogene and therapeutic target [96]. However, SIRTs remain difficult to block therapeutically due their bivalent nature as a repressor and tumor promoter [97].

The methylation of histones, catalyzed by lysine methyltransferases (KMTs) and reversed by demethylases (KDMs), also plays a role in gene regulation in tumor biology (Figure 1B). In general, specific methylation marks, such as H3K4me3 and H3K36me3, are associated with transcriptional activation, while H3K27me3 and H3K9me3 are related to gene repression [93]. A recent study highlighted how KMT2D, an H3K4me1/2 methyltransferase, acts as an onco-suppressor in melanoma. The loss of KMT2D results in a remodeling of the enhancer regions (decrease in H3K4me1 on specific regulatory regions) involved in the regulation of glycolytic and IGF1R-AKT pathways, generating a direct metabolic impact. The authors of this study confirmed in cellular and mouse models how KMT2D-deficient tumors become dependent on glycolysis and more sensitive to anti-glycolysis and anti-IGF1R drugs, opening up new treatment opportunities [98,99]. Vogel et al. showed that the overexpression of KDM5B, H3K4 demethylase, also known as JARID1b, in melanoma cells induces a profound metabolome remodeling characterized by drug resistance and the OXPHOS-dependent phenotype. In parallel, pharmacological inhibition or knockdown of KDM5B leads to a reduction in the GSH/GSSG ratio, NADPH production, and proliferation in 3D models, suggesting a direct impact on metabolic fitness and tumor invasiveness [100].

### 3.3. Non-Coding RNAs

Noncoding RNAs (ncRNAs) are RNA molecules that cover almost the entire genome as opposed to coding regions. They can be grouped into three major functional and structural classes: microRNAs (miRNAs), long non-coding RNAs (lncRNAs), and circular RNAs (circRNAs). Over time, a plethora of functions performed by these molecules have been discovered, as follows: (i) transcriptional and post-transcriptional gene expression regulation; (ii) epigenetic regulation; (iii) Alternative Splicing; (iv) Molecular Decoy; and (v) Scaffold, etc. [101]. ncRNAs are regulatory molecules that act either as oncogenes or tumor suppressors in cancer. They influence cancer development and progression, modulating proliferation, apoptosis, invasion, and metastasis [102]. There are many studies based on the role of ncRNAs in melanoma and its metabolic reprogram ability [103] (Table 1).

The H19 lncRNA acts on the HIF-1α/E2F3 axis by up-regulating the glycolytic pathway [104,112]. Similarly, the LINC00518 and CCHE1 lncRNAs boost the glycolysis: firstly, acting as a sponge for miR-33a-3p, they reduce its inhibition on HIF-1α, enhancing the HIF-1α/LDHA axis by increasing tumor radioresistance [105]; in the latter case, CCHE1-driven metabolic reprogramming supports the FGFR1-LDHA complex to increase glycolysis, which in turn promotes melanoma progression and chemoresistance [106]. circMYC (hsa_circ_0085533) is markedly up-regulated in melanoma tissues and promotes tumor cell proliferation, driving glycolytic flux; sponging off the tumor suppressor miR-1236, circMYC relieves LDHA repression, leading to increased rates of extracellular acidification and lactate production [107]. In contrast, miR-33a-3p and miR-33a act as oncosuppressors, binding to the 3′UTR of HIF-1α, suppressing its post-transcriptional expression, and inhibiting proliferation and glycolysis in melanoma cells [108,109].

ncRNAs, in a dynamic epigenetic context, drive metabolic plasticity in melanoma post-therapy, activating the metabolic switch in response to drug treatment. For example, the silencing of SAMMSON, an SOX10-regulated lncRNA present in more than 90% of melanomas, not only impairs mitochondrial homeostasis by inducing apoptosis but also potentiates the efficacy of BRAF and MEK inhibitors by hindering the ability of cells to adapt metabolically to the MAPK pathway blockade [110]. Similarly, LENOX (LINC00518) is up-regulated following MAPK inhibition and promotes, via interaction with RAP2C and DRP1, a switch from glycolysis to OXPHOS that confers drug resistance [111].

## 4. Metabolites as Epigenetic Modulators in Melanoma Treatment

Metabolites are not simply metabolism products, but cofactors and (co)substrates essential for the catalytic activity of most epigenetic enzymes. In the melanoma context, plasma and intracellular concentrations of acetyl-CoA, glutamine, alpha ketoglutarate (α-KG), lactate, etc., dynamically reshape themselves in response to changes in nutrients, cellular stress conditions, the microenvironment, and response to treatments. These fluctuations not only redirect cellular metabolism but also rewrite epigenomic landscapes, shaping cancer cell behavior and influencing therapeutic resistance [113,114] (Figure 2).

### 4.1. Acetyl-CoA

Acetyl-CoA represents a crucial metabolic node that connects several biochemical circuits such as glycolysis, TCA, lipogenesis, and epigenetic modifications. Tumor cells specifically exploit acetyl-CoA to support de novo lipid synthesis and to promote the ketogenesis pathway. The uncontrolled increase in ketone bodies in cancer cells is known to promote tumor growth in different contexts, such as melanoma. Xia and collaborates showed that in BRAF^V600E^ melanomas, the condensation of two acetyl CoA molecules into acetoacetate results in the activation of MAPK signaling and, consequently, tumor growth [115]. HAT enzymatic activity is highly sensitive to intracellular acetyl-CoA concentrations. Indeed, Acetyl-CoA availability serves as a limiting factor for histone acetylation, particularly under nutrient-restricted or stressed conditions [116]. Mechanistically, Wellen et al. demonstrated that glucose-derived citrate, via ACLY, fuels nuclear acetyl-CoA pools that support p300-mediated histone acetylation in proliferating cells [117]. Similarly, Cluntun et al. showed that mitochondrial–cytosolic acetyl-CoA shuttling regulates H3 acetylation and affects the transcription of growth-related genes in cancer [118].

ACLY enzymes are overexpressed in melanoma and promote tumor growth by the activation of the MITF-PGC1α axis, essential for mitochondrial biogenesis. ACLY promotes histones’ acetylation in the MITF promoter via acetyltransferase P300, inducing mitochondrial oxidation, adaptive resistance to MAPK inhibitors, and subsequent tumor proliferation. Its inhibition, leading to the modification of acetyl-CoA levels, could synergize with MAPK inhibitors, offering a promising combination approach in melanoma therapy [50,119,120]. Recent evidence has revealed a direct link between acetyl-CoA metabolism and the regulation of ICIs in cancer [121,122]. In particular, Wang et al. demonstrated how in melanoma, increased acetyl-CoA, acting as a co-substrate for acetyltransferase P300, induces the acetylation of residue H3K27 in the PD-L1 promoter, leading to its overexpression (Figure 2). Again, the genetic or pharmacological depletion of ACLY significantly reduces PD-L1 expression, increases CD8^+^ T lymphocyte infiltration, and potentiates the response to anti-PD-1 immunotherapy. In addition, the co-administration of ACLY inhibitors and immunotherapy showed synergistic efficacy in slowing tumor growth [52]. This research identifies acetyl-CoA and its key enzyme ACYL as promising metabolic–epigenetic targets whose inhibition could be strategically integrated into therapeutic regimens, opening the way for new approaches to overcome resistance in melanoma.

### 4.2. α-KG and Glutamine

α-KG, a key TCA cycle intermediate, is a cofactor for epigenetic enzymes, such as histone demethylases containing the Jumonji domain (JHDM) and TETs involved in DNA demethylation [123]. TET activity is highly dependent on α-KG as a co-substrate and it is competitively inhibited by 2-hydroxyglutarate (2-HG), an oncometabolite produced by the mutant IDH1/2 in other tumor types [124]. The loss of 5hmC, a hallmark of TET inactivation, is a recurrent epigenetic feature in melanoma and correlates with tumor progression and stemness. The restoration of TET2 or IDH2 activity in melanoma cells re-establishes the 5hmC landscape and suppresses tumor growth in vitro and in vivo, highlighting the therapeutic potential of modulating TET activity [125]. Melanoma cells often are resident in glutamine-poor microenvironments; this condition induces the reduced production of glutamate, an essential substrate for α-KG synthesis. Decreased α-KG production results in histone hypermethylation, tumor de-differentiation, and resistance to therapy. A recent study by Gabra and collaborators demonstrated how dietary glutamine supplementation reduced tumor growth and increased sensitivity to targeted therapies. The study showed that the intra-tumoral increase in α-KG levels attenuated H3K4 tri-methylation, inactivating many melanoma oncogenic genes (MITF, AXL, CD271, etc.) and phosphatidylinositol 3-kinase (PI3K) and MAPK pathways, independently of the genetic background of the tumor [126]. However, the tumor glutamine role remains debated. Several studies showed as in solid tumors that the cells are “glutamine-dependent”, strongly competing with the immune cells of the microenvironment. A “tug of war” of glutamine is established between the tumor and immune cells. From this evidence grows the idea of blocking glutamine intake or conversion in cancer cells to restore glutamine to immune cells and reverse the resistance phenomena [127,128]. In the melanoma field, a recent study has further highlighted how the availability of glutamine-derived α-KG not only influences tumors’ intrinsic features but also profoundly shapes tumor and immune cells’ interactions in an epigenetic manner. Indeed, exogenous α-KG increases ten-eleven translocation 2/3 (TET2/3) activity by increasing 5-hydroxymethylcytosine (5-hmC) at the Programmed Death-Ligand 1 (PD-L1) promoter. This modification stabilizes Signal Transducer and Activator of Transcription 1/3 (STAT1/3) binding, increasing PD-L1 expression. The combination of α-KG supplementation with anti-PD-1 therapy results in improved therapeutic efficacy; indeed, α-KG-induced upregulation of PD-L1 increases tumor susceptibility to the immune checkpoint blockade, restoring and enhancing cytotoxic T-cell activity [129]. α-KG and succinyl-CoA metabolism interaction adds another layer of complexity. Liang et al. showed that the α-KG analogs (DMKs) could increase intracellular succinyl-CoA and, consequently, PD-L1 succinylation and lysosomal degradation. This post-translational regulation of PD-L1 is correlated with increased CD8^+^ T cell activation and tumor rejection [130].

All this evidence underlines how, in melanoma, glutamine and α-KG are not only simple metabolic intermediates but key modulators between cell metabolism, epigenetic phenomena, and treatment response. α-KG enhances TET enzyme activity and fosters a more immunoreactive microenvironment. As shown in Figure 2, α-KG modulates tumor–immune cell interactions and improves response to anti-PD-1 therapy.

### 4.3. Lactate

In the recent years, the histone modification, lactylation, emerged for its critical role on tumorigenesis and resistance processes [131,132,133]. Zhang et al., in 2019 [134], defined lactylation as a novel post-translational modification of histones, in which the lactyl group binds to lysine residues. Traditionally, lactate produced by the aerobic glycolysis typical of cancer cells has been considered a simple metabolic “waste” associated with progression and immunosuppression. Several studies, however, have shown that excess lactate can also act as an epigenetic “messenger,” affecting both the tumor and the surrounding microenvironment. Indeed, lactate and lactylation have been implicated in promoting tumor progression in colon cancer, clear cell renal cell carcinoma, prostate cancer, and hepatocellular carcinoma, by mechanisms including immune escape, the activation of oncogenes, angiogenesis, and the establishment of an immunosuppressive microenvironment [134].

Recent studies have demonstrated that lactate acts as a context-dependent epigenetic regulator. In uveal melanoma, lactate-mediated histone H3K18 lactylation promotes chromatin relaxation and the upregulation of OXPHOS-related genes, thereby inducing a quiescent, low-proliferative cellular phenotype [135]. Another pro-tumor epigenetic circuit, induced by lactate accumulation, promotes the activation of YTH domain family member 2 (YTHDF2) expression and induces tumor suppressors’ degradation, such as period circadian protein homolog 1 (PER1) and tumor protein 53 (TP53) [136].

In therapy-resistant cutaneous melanoma, re-established glycolysis and lactate production promote the lysine lactylation of lysine-specific demethylase 1 (LSD1), stabilizing its interaction with FOS-like antigen 1 (FosL1). This process represses ferroptosis through transferrin receptor (TFRC) down-regulation, supporting resistance to BRAFi/MEKi and immune escape [137]. Parallel computational analyses in cutaneous melanoma have identified lactylation-associated gene signatures, specifically calmodulin-like protein 5 (CALML5), involved in patient prognosis and immune cell infiltration; these signatures lead to the development of lactate–tumor microenvironment (LAC-TME) classifiers, with potential clinical utility for immunotherapy stratification [138].

As illustrated in Figure 2, these findings highlight the multifaceted role of lactate as a metabolic and epigenetic modulator and demonstrated its implication in therapy resistance and immune dynamics through distinct but interconnected molecular mechanisms.

## 5. Tumor Microenvironment and the Epigenetic–Metabolic Axis in Melanoma

TME in melanoma is a dynamic and metabolically active compartment that profoundly influences tumor behavior, therapy response, and immune evasion [139,140].

One of the most prominent metabolic features of melanoma TME is hypoxia, that stabilizes HIFs and promotes the glycolytic switch. This process enhances lactate production and export through the coordinated activity of key enzymes and transporters involved in lactate metabolism, including LDHA and MCT1-4 transporters. Concomitantly, the acidification of the TME influences the availability and functionality of metabolic cofactors that serve as substrates or regulators of chromatin-modifying enzymes. For example, acidic conditions suppress intracellular levels of α-KG. The inhibition of these enzymes favors the persistence of repressive methylation marks, contributing to the silencing of tumor suppressor genes and fostering a dedifferentiated and therapy-resistant phenotype [141,142].

Tumor-associated macrophages (TAMs), one of the most abundant immune populations in the melanoma TME, exemplify this metabolite-driven reprogramming [143]. Indeed, the lactate secreted by melanoma cells induces histone lactylation in TAMs. Macrophages accumulate H3K18 lactylation, inducing Arg1 transcription and other M2 markers [132]. Lactate also fuels mitochondrial metabolism in Inteleukin-4 (IL-4)-polarized TAMs. Its oxidation to citrate, which is then cleaved by ACLY to generate acetyl-CoA, leads to histone acetylation in M2 gene promoters. This process promotes the expression of immunosuppressive genes such as arginase-1 (Arg1) and vascular endothelial growth factor a VEGFa, supporting an M2-like protumor phenotype [144].

Cancer-associated fibroblasts (CAFs) are a pivotal component of melanoma TME, and their dynamic crosstalk with cancer cells is increasingly recognized as an important driver of disease progression [145]. CAFs contribute to tumor progression through the secretion of soluble factors and metabolic intermediates that reprogram surrounding cells. [140]. Pro-inflammatory cytokines such as IL-6 and CXCL5, derived from CAFs, activate signaling cascades including JAK/STAT3 and PI3K/AKT, promoting epigenetic remodeling through DNMT1 and H3K27ac deposition. In parallel, tumor-derived metabolites, such as lactate, can drive α-KG production and TET activation in mesenchymal stromal cells, sustaining a tumor-supportive CAF phenotype [146]. In melanoma, tumor-derived inflammatory cytokines activate melanoma-associated fibroblasts (MAFs), promoting metabolic and functional reprogramming. This includes increased IL-6 production under hypoxic conditions and the release of immunomodulatory mediators. The resulting lactate-enriched and immunosuppressive microenvironment impairs CD8^+^ T cell cytotoxicity and contributes to immune evasion mechanisms [147].

## 6. Therapeutic Strategies and Clinical Trials Targeting the Metabolic–Epigenetic Axis in Melanoma

Melanoma cells escaping BRAF/MEKi or ICIs frequently exhibit altered metabolic states that impact chromatin accessibility, histone modifications, and drug tolerance. This interplay between metabolism and epigenetics promoted the development of combination strategies aimed at overcoming therapeutic resistance. Several clinical trials reveal that strategies targeting only one component, either the epigenetic or metabolic axis, frequently result in suboptimal efficacy (Table 2 [148]).

For example, epigenetic monotherapies or in combination with targets/ICIs, based on DNMT, HDAC, or EZH2 inhibitors (e.g., decitabine-based [NCT01876641], CPI-1205-based [NCT03525795], and entinostat/pembrolizumab study [NCT02437136]), often fail to yield durable responses. However, it is important to note that the trial investigating guadecitabine plus ipilimumab [NCT02608437] demonstrated encouraging long-term responses in a subset of patients, especially those with pre-existing immune-active signatures, highlighting the potential of epigenetic priming when appropriately combined and biologically stratified [149].

Similarly, metabolism-targeting approaches have produced limited success in melanoma. For instance, telaglenastat [NCT02771626], a glutaminase inhibitor, demonstrated a favorable safety profile in a phase I/II study, but showed only modest response rates and no significant clinical benefit in patients with advanced disease [153]. In the case of phenformin, the Mitochondrial Complex I inhibitor, evaluated in combination with dabrafenib and trametinib, some objective responses were observed, particularly in patients naïve to MAPK inhibition, yet the overall clinical benefit was modest. Furthermore, treatment was associated with notable safety concerns, including instances of reversible lactic acidosis, limiting its broader applicability in unselected melanoma populations [152]. Another example is the trial of AZD3965, an MCT1 inhibitor (NCT01791595), which was suspended following a case of life-threatening hyperlactaemic acidosis in a patient with metastatic melanoma. Despite a mechanistically sound rationale, blocking lactate export in tumors reliant on aerobic glycolysis, the treatment unmasked a severe “hyper-Warburgism” state driven by excessive tumor glucose uptake and lactate production. This case underscores the complexity of applying metabolic inhibitors in the clinic, particularly in patients with a high glycolytic tumor burden, and emphasizes the need for careful metabolic profiling before patient enrollment [154].

This pattern was clear with the clinical failure of the IDO1 inhibitor Epacadostat. In the ECHO-301 trial [NCT02752074], where Epacadostat combined with pembrolizumab, it did not outperform pembrolizumab monotherapy in metastatic melanoma (subsequently observed in sarcomas, non-small cell lung cancer, renal cell carcinoma, and urothelial carcinoma). Together, these observations emphasize that isolated or dual blockade strategies, without a comprehensive integration of the metabolic, epigenetic, and immune axes, are insufficient to overcome the plasticity and immune evasion typical of melanoma [155,157].

Despite these delays, notable successes in other cancer contexts illustrate the potential of integrated metabolic–epigenetic targeting. In melanoma, dysregulated lipid metabolism, driven by SCD1 overexpression, was shown to blunt the sensitivity to EZH2 inhibitors; the pharmacological inhibition of SCD1 successfully restored drug sensitivity in in vitro and in vivo models. Mechanistically, SCD1 overexpression increased MUFA synthesis, which interferes with the PRC2 complex function by altering membrane composition and S-adenosylmethionine (SAM) availability, leading to impaired H3K27me3 deposition at key tumor suppressor genes. Inhibiting SCD1 re-establishes proper EZH2-mediated transcriptional repression, thereby restoring epigenetic control and sensitizing melanoma cells to EZH2-targeted therapies [158].

In isocitrate dehydrogenase 1 (IDH1)-mutant Acute Myeloid Leukemia (AML), the combination of Ivosidenib with the hypomethylating agent azacitidine, significantly improved clinical outcomes compared to monotherapy. Similarly, DNMT or HDAC inhibitors with the B-cell leukemia/lymphoma 2 (BCL2) inhibitor Venetoclax has proven highly effective in both hematological malignancies and solid tumors (liver, lung, colon, breast). In glioblastoma, the combination of Panobinostat and the FAO inhibitor etomoxir leveraged metabolic rewiring to enhance therapeutic efficacy [159].

The therapeutic relevance of targeting the metabolic–epigenetic axis in melanoma lies in its dual capacity to reprogram tumor intrinsic transcriptional states and remodel the immunosuppressive tumor microenvironment. Achieving future clinical success will require the rational integration of metabolic and epigenetic agents. This must be supported by biomarker-guided patient stratification and high-throughput technologies capable of identifying actionable vulnerabilities.

## 7. Conclusions and Future Prospective

Melanoma is an extremely heterogeneous tumor characterized by profound metabolic and epigenetic plasticity that complicates therapeutic approaches and promotes resistance mechanisms. Metabolism and epigenetics crosstalk represent a central node in the cell state transition regulation, supporting tumor adaptation to environmental and therapeutic pressures. In this review, we have dissected the principal mechanisms by metabolism that shapes the epigenome and vice versa, emphasizing how fluctuations in metabolites such as acetyl-CoA, α-KG, and lactate act as direct modulators of chromatin structure and gene expression. However, several challenges remain. The intrinsic genetic and epigenetic melanoma heterogeneity demands highly personalized therapeutic approaches. In addition, metabolic/epigenetic interventions need to consider tumor plasticity and microenvironmental limitations to prevent rapid adaptation and relapse. In conclusion, targeting the metabolic–epigenetic interface represents a promising, yet complex, therapeutic strategy for melanoma and other highly plastic cancers. Future efforts should focus on developing precise strategies able to reprogram tumor cell fate, enhance immune responses, and prevent or reverse therapy resistance. A deep understanding of the bidirectional interplay between metabolism and epigenetic phenomena will be critical to designing the next-generation cancer treatments. Due to the advent of new high-throughput technologies, including metabolomics, single-cell omics, and CRISPRi-based functional screenings, it is now possible to deeply interrogate the metabolic–epigenetic landscape of tumors, uncover actionable vulnerabilities, and rationally design combinatorial therapeutic approaches that could redefine the clinical management of melanoma and other highly plastic cancers.

## Figures and Tables

**Figure 1 pharmaceuticals-18-00853-f001:**
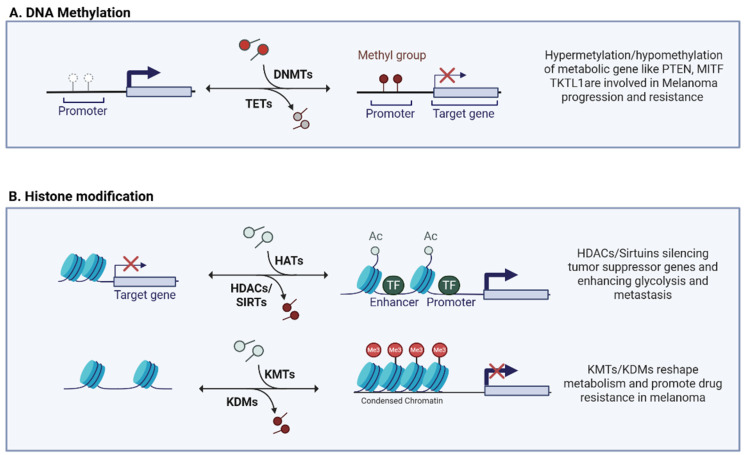
(**A**,**B**) Schematic representation of epigenetic mechanisms involved in melanoma metabolic reprogramming: DNA methylation; histone modifications. Created in BioRender. Giuliani, S. (2025) https://app.biorender.com/illustrations/680f7fb6830216ca4c9e5fdc.

**Figure 2 pharmaceuticals-18-00853-f002:**
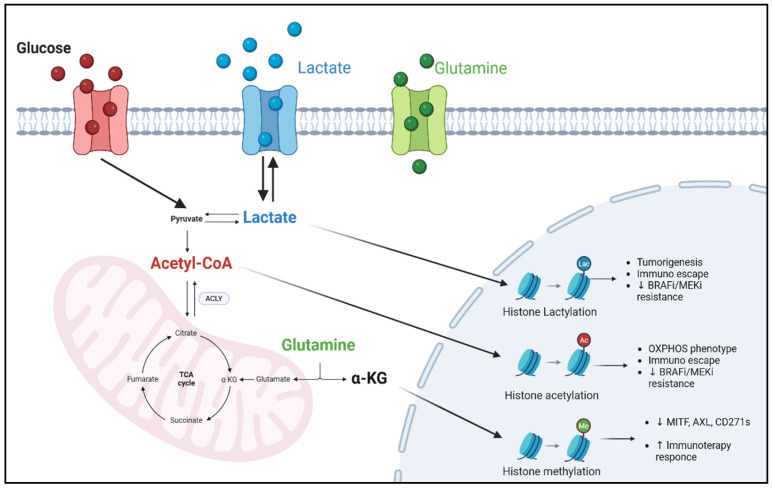
Overview of principal metabolites acting as cofactors and substrates for epigenetic enzymes in melanoma. Acetyl-CoA, α-KG, and lactate dynamically modulate chromatin states through histone acetylation, DNA/histone demethylation, and histone lactylation, respectively. Acetyl-CoA promotes H3K27 acetylation via p300 at the PD-L1 promoter; α-KG activates TET2/3 and histone demethylases, enhancing immune visibility; lactate induces H3K18 lactylation, promoting immune evasion or quiescence. Created in BioRender. Giuliani, S. (2025) https://app.biorender.com/illustrations/680f4315e7e7d75d587258ed.

**Table 1 pharmaceuticals-18-00853-t001:** ncRNAs involved in melanoma metabolic reprogramming.

ncRNA	Effect	Citation
H19	Upregulates glycolysis via HIF-1α/E2F3 axis	[104]
LINC00518	Enhances glycolysis by sponging miR-33a-3p, promoting HIF-1α/LDHA axis and increasing radioresistance	[105]
CCHE1	Promotes glycolysis through FGFR1-LDHA complex, supporting melanoma progression and chemoresistance	[106]
circMYC (hsa_circ_0085533)	Increases glycolytic flux by sponging miR-1236, relieving LDHA repression	[107]
miR-33a	Acts as tumor suppressor by binding HIF-1α 3′UTR, inhibiting proliferation and glycolysis	[108,109]
SAMMSON	Impairs mitochondrial homeostasis, increases sensitivity to BRAF/MEK inhibitors by blocking metabolic adaptation	[110]
LENOX (LINC00518)	Promotes switch from glycolysis to OXPHOS following MAPK inhibition, contributing to drug resistance	[111]

**Table 2 pharmaceuticals-18-00853-t002:** Summary of clinical trials investigating metabolic and epigenetic combination therapies in melanoma.

Drugs	Target(s)	Mechanism of Action	Clinical Status	Cancer Type	Clinical Trial ID
Guadecitabine + Ipilimumab	DNMT1; CTLA-4	Hypomethylating agent boosts tumor immunogenicity; CTLA-4 blockade activates T-cell response	Phase I completed	Metastatic melanoma	NCT02608437 Guadecitabine [149]
Decitabine + Vemurafenib	DNMT1; BRAF^V600E^	DNA demethylation delays MAPK-inhibitor resistance via epigenetic reprogramming	Phase Ib completed	BRAF-mutant melanoma	NCT01876641 [150]
Entinostat + Pembrolizumab	Class I HDAC; PD-1	HDAC inhibition reduces MDSCs and increases antigen expression; potentiates PD-1 checkpoint blockade	Phase II (ENCORE-601)	Anti–PD-1–refractory melanoma	NCT02437136 [151]
CPI-1205 + Ipilimumab	EZH2; CTLA-4	Reverses gene silencing via EZH2 inhibition; enhances tumor visibility and immune responsiveness	Phase I/II completed	Solid tumors incl. melanoma	NCT03525795
Phenformin + Dabrafenib/Trametinib	Mitochondrial Complex I; BRAF/MEK	Biguanide blocks OXPHOS in resistant cells; MAPK inhibitors target glycolysis-reliant cells	Phase I completed (RP2D defined)	BRAF^V600E^ melanoma	NCT03026517 [152]
Telaglenastat (CB-839) + Nivolumab	Glutaminase (GLS1); PD-1	Glutamine metabolism inhibition reshapes redox and TME to increase T-cell effectiveness	Phase I/II completed	Advanced melanoma	NCT02771626 [153]
AZD3965	MCT1	Inhibits lactate export in glycolytic tumors	Phase I (suspended in melanoma)	Solid tumors and lymphomas	NCT01791595 [154]
INCB059872 + Epacadostat + Pembrolizumab	LSD1; IDO1; PD-1	LSD1 inhibitor enhances tumor immunogenicity; IDO1 inhibition reduces immunosuppression; PD-1 blockade activates T-cell response	Phase I/II completed	Advanced solid tumors incl. melanoma	NCT02959437 [155]
Epacadostat + Pembrolizumab	IDO1; PD-1	IDO1 inhibition to enhance PD-1 blockade efficacy in unselected melanoma patients	Phase III completed	Metastatic melanoma	NCT02752074 [156]

## Data Availability

No new data were created or analyzed in this study. Data sharing is not applicable to this article.

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
