# Peer review of "Metabolic Reprogramming in Melanoma: An Epigenetic Point of View"

_pharmaceuticals, 2025, doi:10.3390/ph18060853_

Round 1
Reviewer 1 Report
Comments and Suggestions for Authors
- The review lacks stating the main hypothesis clearly. The authors should define the main objective of the review in more details in the introduction and conclusion. For example, the authors could summarize the key challenges in current melanoma therapy and directing this review as a guide to overcome these challenges through metabolic-epigenetic interface.
-
Although the review presents high-quality figures, some figures are still not sufficiently integrated within the text. For example, Figure 2 on demonstrating metabolite-epigenetic interactions is mentioned briefly without in detail explanation on how each metabolite specifically affects chromatin architecture in melanoma.
-
Authors should expand figure legends to include more details and refer to them deeply within the main text.
-
The authors could use subheadings to link visual illustrations with with text (e.g. Figure 2 illustrates how α-KG modulates TET activity in response to therapy ..... and so on).
-
Although the review discusses some preclinical findings, it should present deeper examination of clinical discussion. For example, how ACLY or SCD1 targeting could affect patient stratification or combination therapy regimens with current BRAF/MEK inhibitors. This should be more explained in tables for example.
Recommendation: Include a table summarizing potential epigenetic-metabolic targets (e.g., ACLY, KDM5B, SIRT6) with their associated inhibitors, clinical status (preclinical/clinical), and therapeutic potential in melanoma—referencing current trials from databases such as ClinicalTrials.gov.
Author Response
Response to Reviewer 1 Comments
- Summary
"Thank you for dedicating your time to review our manuscript. Below we provide detailed responses to your comments, and the corresponding revisions have been clearly highlighted in the resubmitted documents."
- Point-by-point response to Comments and Suggestions for Authors
Comments 1: The review lacks stating the main hypothesis clearly. The authors should define the main objective of the review in more details in the introduction and conclusion. For example, the authors could summarize the key challenges in current melanoma therapy and directing this review as a guide to overcome these challenges through metabolic-epigenetic interface.
Response 1: Thank you for pointing this out. We agree with this comment. Therefore, we have revised the Introduction to better define the main objective of the review by explicitly outlining the current therapeutic challenges in melanoma. Specifically, we have added the sentence:
“In the current therapeutic landscape, immune checkpoint inhibitors (ICIs) targeting PD-1 and CTLA-4, as well as targeted therapies against MAPK pathway components (e.g., BRAF and MEK inhibitors), have become standard-of-care treatments in melanoma. Nonetheless, recurrence remains a significant clinical challenge, with relapse rates ranging from 20–40% in initially responsive patients, due to the emergence of resistance mechanisms and tumor plasticity.”
“In light of this, we hypothesize that the bi-directional interface between metabolism and epigenetics represents a crucial node underlying melanoma's high adaptive capacity and resistance to therapies,” thereby reinforcing the central hypothesis and objective of the review.
Furthermore, to strengthen the translational relevance of our analysis, we have revised the final section of the manuscript by adding a dedicated paragraph that discusses ongoing and completed clinical trials involving metabolic and epigenetic drugs in melanoma. This aims to bridge the gap between mechanistic insights and therapeutic applications, in line with the review’s goal to guide future treatment strategies through the metabo-epigenetic interface.
Comments 2-3-4:
- Although the review presents high-quality figures, some figures are still not sufficiently integrated within the text. For example, Figure 2 on demonstrating metabolite-epigenetic interactions is mentioned briefly without in detail explanation on how each metabolite specifically affects chromatin architecture in melanoma.
- Authors should expand figure legends to include more details and refer to them deeply within the main text.
- The authors could use subheadings to link visual illustrations with with text (e.g. Figure 2 illustrates how α-KG modulates TET activity in response to therapy ..... and so on).
Response: Thank you for these valuable suggestions. We agree with the reviewers’ comments. In response, we have revised Figure 2 legend: “Overview of principal metabolites acting as cofactors and substrates for epigenetic enzymes in melanoma. Acetyl-CoA, α-KG and lactate dynamically modulate chromatin states through histone acetylation, DNA/histone demethylation and histone lactylation, respectively. Acetyl-CoA pro-motes H3K27 acetylation via p300 at the PD-L1 promoter; α-KG activates TET2/3 and histone demethylases enhancing immune visibility; lactate induces H3K18 lactylation, promoting immune evasion or quiescence. This image was created with BioRender (https://biorender.com).” to provide a more detailed and explicit description of the key metabolic intermediates and their respective roles in modulating epigenetic enzymes and chromatin architecture in melanoma.
In addition, we have enhanced the integration of Figure 2 within the main text by inserting specific references to the figure in Sections 4.1, 4.2, and 4.3, where the roles of metabolite are discussed. These references are intended to create a clearer link between the textual content and the visual illustration, as suggested.
Moreover, we have revised the relevant paragraphs to more thoroughly describe how each metabolite interacts with epigenetic regulators thereby influencing chromatin accessibility and gene expression programs that contribute to melanoma progression and therapy resistance.
In Section 4.1, we added the following:
“HAT enzymatic activity is highly sensitive to intracellular acetyl-CoA concentrations. Indeed, acetyl-CoA availability serves as a limiting factor for histone acetylation, particularly under nutrient-restricted or stressed conditions [117]. Mechanistically, Wellen et al. demonstrated that glucose-derived citrate, via ACLY, fuels nuclear acetyl-CoA pools that support p300-mediated histone acetylation in proliferating cells [118]. Similarly, Cluntun et al. showed that mitochondrial-cytosolic acetyl-CoA shuttling regulates H3 acetylation and affects transcription of growth-related genes in cancer [119].”
In Section 4.2, we incorporated a more detailed explanation of α-KG and TET activity:
“TET activity is highly dependent on α-KG as a co-substrate and it is competitively inhibited by 2-hydroxyglutarate (2-HG), an oncometabolite produced by mutant IDH1/2 in other tumor types [125]. Loss of 5hmC, a hallmark of TET inactivation, is a recurrent epigenetic feature in melanoma and correlates with tumor progression and stemness. Restoration of TET2 or IDH2 activity in melanoma cells re-establishes the 5hmC landscape and suppresses tumor growth in vitro and in vivo, highlighting the therapeutic potential of modulating TET activity [126].”
These changes collectively improve the coherence between the figure and the narrative, enhancing the clarity and educational value of the manuscript.
Comments 5: Although the review discusses some preclinical findings, it should present deeper examination of clinical discussion. For example, how ACLY or SCD1 targeting could affect patient stratification or combination therapy regimens with current BRAF/MEK inhibitors. This should be more explained in tables for example.
Response: Thank you for this insightful comment. We agree with the reviewer that a deeper clinical perspective would strengthen the manuscript. In response, we have added a new Section 6 titled “Therapeutic Strategies and Clinical Trials Targeting the Metabolic-Epigenetic Axis in Melanoma”. This section provides a comprehensive overview of ongoing and completed clinical trials either as monotherapies or in combination with current standard treatments.
We have also included a summary table that highlights relevant compounds, their molecular targets, mechanisms of action, clinical status, cancer type, and potential therapeutic rationale. This addition not only reinforces the translational relevance of the molecular mechanisms described earlier in the review but also offers a clearer and more integrative view of how targeting the metabolic-epigenetic interface could inform patient stratification and combination therapy design.

Reviewer 2 Report
Comments and Suggestions for Authors
Giuliani and colleagues present a review about the metabolic and epigenetic interaction in melanoma cancer cells. The topic is of high clinical relevance because innovative therapeutic approaches are based on these novel insights. The manuscript is written in a comprehensive and clear manner, the complex and dynamic theme is presented in a vivid order.
Although the authors focus on melanoma the generalizability of some metabolic or epigenetic mechanisms across cancer types are mentioned and clinical translation is finally discussed.
The reviewer has few issues of concern.
Major comments:
The fundamental observation of Otto Warburg (and Karl Posener und Erwin Negelein) was first published 1924 (not 1956): O. Warburg, K. Posener, E. Negelein, Über den Stoffwechsel der Carcinomzelle, Biochem Zeitschr. 152 (1924) 309–344 (line 62).
The authors do not mention the role pyruvatkinase M1 and in particular M2, a “key player” of the Warburg effect (see Hamanaka et al.; Science; Nov 2011) which also plays an important role in epigenetic modification such as histone phosphorylation (e.g. Yang et al., Cell, Vol 150, 2012). PKM2 is of particular interest in melanoma treatment, Zhou et al. suggest Benserazide as novel treatmnent approach (Int J of Cancer, 2020; PMID: 31652354).
„The researchers link PD-L1 upregulation to enhanced CD8+ and CD4+ T cells cytotoxicity“ (line 398) – is this indeed true? PD-L1 upregulation results in T cell silencing – which can be reverted by blocking antibodies.
Minor comments:
- Please check the sentence: “Metabolic plasticity, enabling melanoma to adapt more quickly and effectively to external stimuli.” (line 36)
- Spell error “transcrition“ (line 90)
Author Response
Response to Reviewer 2 Comments
- Summary
"Thank you for dedicating your time to review our manuscript. Below we provide detailed responses to your comments, and the corresponding revisions have been clearly highlighted in the resubmitted documents."
- Point-by-point response to Comments and Suggestions for Authors
Comment 1:
The fundamental observation of Otto Warburg (and Karl Posener und Erwin Negelein) was first published 1924 (not 1956): O. Warburg, K. Posener, E. Negelein, Über den Stoffwechsel der Carcinomzelle, Biochem Zeitschr. 152 (1924) 309–344 (line 62).
Response 1:
Thank you for this correction. We have updated the historical reference accordingly. The correct citation is now included in Reference 14, and the corresponding date has been revised in the text (page X, line Y).
Comment 2:
The authors do not mention the role of pyruvate kinase M1 and in particular M2, a “key player” of the Warburg effect (see Hamanaka et al., Science, 2011), which also plays an important role in epigenetic modification such as histone phosphorylation (Yang et al., Cell, 2012). PKM2 is of particular interest in melanoma treatment; Zhou et al. suggest benserazide as novel treatment (Int J Cancer, 2020; PMID: 31652354).
Response 2:
As suggested by the reviewer, we have integrated a dedicated passage on the role of pyruvate kinase isoforms PKM1 and PKM2 into the section on glycolytic metabolism. We discuss their dual function in metabolic regulation and epigenetic modulation, including the non-canonical role of PKM2 in histone H3-T11 phosphorylation and its implication in gene transcription and tumorigenesis. We have also cited the potential therapeutic role of benserazide as proposed in Zhou et al. (PMID: 31652354).
Specifically, we have added the sentence: “Another key protein in glycolytic metabolism is pyruvate kinase M2 (PKM2), a regula-tory isoenzyme that catalyzes the last step of glycolysis. PKM2 is up-regulated and al-lows cancer cells to direct metabolites to biosynthesis [29]. Furthemore, PKM2 not only promotes proliferation, but is also involved in non-canonical epigenetic functions, in-cluding histones phosphorylation (e.g., H3-T11), influencing gene expression in re-sponse to metabolic stimuli [30]. “
In melanoma, pharmacological inhibition of PKM2 by Benserazide has been shown to inhibit tumor growth and enhance the efficacy of BRAFi, suggesting possible combined use in target therapy [31].”
Comment 3:
“The researchers link PD-L1 upregulation to enhanced CD8+ and CD4+ T cells cytotoxicity” (line 398) – is this indeed true? PD-L1 upregulation results in T cell silencing – which can be reverted by blocking antibodies.
Response 3:
We appreciate this observation. The original sentence was indeed misleading. We have now rephrased it to clarify that mechanism. The revised sentence: “The combination of α-KG supplementation with anti-PD1 therapy results in improved therapeutic efficacy; indeed, α-KG-induced upregulation of PD-L1 increases tumor susceptibility to immune checkpoint blockade, restoring and enhancing cytotoxic T-cell activity”
Reviewer 2 – Minor Comments
Comment 1:
Please check the sentence: “Metabolic plasticity, enabling melanoma to adapt more quickly and effectively to external stimuli.” (line 36)
Response 1:
Thank you for pointing this out. We have rephrased the sentence to improve clarity and grammar. “The metabolic plasticity of melanoma allows tumor cells to rapidly and efficiently adapt to external stimuli, including therapeutic pressures”
Comment 2:
Spell error “transcrition“ (line 90)
Response 2:
The spelling error has been corrected to “transcription”.

Reviewer 3 Report
Comments and Suggestions for Authors
The paper provides an overview of the metabolic characteristics of melanoma and how the cancer leverages a wide array of metabolic proteins and genes to support its survival and growth. I believe the paper effectively summarizes complex and intricate signaling mechanisms without delving into the detailed pathways of each individual process it describes. It may be worth revisiting the sentences on lines 223 and 469, as they might not be well-formulated. Overall, the paper is informative, covering many well-established facts while also referencing recent literature and presenting findings from newer studies. Therefore, I believe it can be a valuable resource for considering the metabolic plasticity of melanoma, how it is achieved, and why it is important.
Author Response
Response to Reviewer 3 Comments
- Summary
"Thank you for dedicating your time to review our manuscript. Below we provide detailed responses to your comments, and the corresponding revisions have been clearly highlighted in the resubmitted documents."
- Point-by-point response to Comments and Suggestions for Authors
General Comment:
We sincerely thank the reviewer for the positive evaluation and thoughtful feedback. We appreciate your recognition of the manuscript’s relevance in summarizing both established and emerging findings on metabolic plasticity in melanoma. Please find below our specific response.
Comment 1:
It may be worth revisiting the sentences on lines 223 and 469, as they might not be well-formulated.
Response 1:
Thank you for pointing this out. We have carefully revised the sentences in question to improve clarity and readability:
revised version:
“In addition, in vivo studies showed that DNMT1 promotes tumorigenesis in melanoma cells by directly activating the PI3K/AKT/mTOR pathway. This occurs through repression of the interaction between two inhibitory molecules, HSPB8 and BAG3 [85].”
revised version:
“In conclusion, targeting the metabolic–epigenetic interface represents a promising—yet complex—therapeutic strategy for melanoma and other highly plastic cancers.”

Reviewer 4 Report
Comments and Suggestions for Authors
I carefully read and reviewed the paper titled "Metabolic Reprogramming in Melanoma: An Epigenetic point of view". It is a review paper that overviewed metaboepigenetic interconnections in melanoma.
The authors addressed a highly relevant and evolving area of melanoma research: the interplay between metabolic reprogramming and epigenetic modulation, which are central to cancer progression and resistance mechanisms. The review highlighted a two-way relationship: Epigenetic regulation controls metabolic gene expression, and metabolites (like acetyl-CoA, α-KG, SAM) influence epigenetic enzymes.. This interdependence is key for identifying therapeutic vulnerabilities.
However, several issues are noted:
- As a narrative review, the article does not detail a systematic methodology for literature selection or appraisal, which may limit reproducibility and bias transparency.
- The paper well covered broad concepts, but specific molecular pathways (e.g., TET enzyme function, histone acetyltransferase–metabolite interactions, IDH mutations) could be more deeply explored for clarity.
- Authors acknowledged the clinical implications, however there’s a lack of detail on current or ongoing clinical trials, drug candidates, or approved therapies that modulate this axis in melanoma.
- The role of the tumor microenvironment in shaping metabolic and epigenetic changes is absent, despite its growing importance in melanoma biology and therapy resistance.
Author Response
Response to Reviewer 4 Comments
- Summary
"Thank you for dedicating your time to review our manuscript. Below we provide detailed responses to your comments, and the corresponding revisions have been clearly highlighted in the resubmitted documents."
Comment 1:
The paper well covered broad concepts, but specific molecular pathways (e.g., TET enzyme function, histone acetyltransferase–metabolite interactions, IDH mutations) could be more deeply explored for clarity.
Response 1:
We appreciate this suggestion and have addressed it by expanding the mechanistic details in Sections 4.1 and 4.2:
In Section 4.1, we added:
“HAT enzymatic activity is highly sensitive to intracellular acetyl-CoA concentrations. Indeed, acetyl-CoA availability serves as a limiting factor for histone acetylation, particularly under nutrient-restricted or stressed conditions [117]. Mechanistically, Wellen et al. demonstrated that glucose-derived citrate, via ACLY, fuels nuclear acetyl-CoA pools that support p300-mediated histone acetylation in proliferating cells [118]. Similarly, Cluntun et al. showed that mitochondrial-cytosolic acetyl-CoA shuttling regulates H3 acetylation and affects transcription of growth-related genes in cancer [119].”
- In Section 4.2, we incorporated a detailed discussion on α-KG and TET activity:
“TET activity is highly dependent on α-KG as a co-substrate and it is competitively inhibited by 2-hydroxyglutarate (2-HG), an oncometabolite produced by mutant IDH1/2 in other tumor types [125]. Loss of 5hmC, a hallmark of TET inactivation, is a recurrent epigenetic feature in melanoma and correlates with tumor progression and stemness. Restoration of TET2 or IDH2 activity in melanoma cells re-establishes the 5hmC landscape and suppresses tumor growth in vitro and in vivo, highlighting the therapeutic potential of modulating TET activity [126].”
These additions improve the mechanistic depth of the review, in line with the reviewer’s recommendation.
Comment 2:
Authors acknowledged the clinical implications, however there’s a lack of detail on current or ongoing clinical trials, drug candidates, or approved therapies that modulate this axis in melanoma.
Response 2:
We fully agree, and to address this point, we have introduced a new section titled “Therapeutic Strategies and Clinical Trials Targeting the Metabolic–Epigenetic Axis in Melanoma” (Section 6). This section provides an overview of preclinical and clinical efforts to target metabolic and epigenetic vulnerabilities in melanoma, including a summary table that lists compounds, targets, mechanisms of action, and clinical trial identifiers. This addition enhances the translational relevance of the review and provides an up-to-date perspective on therapeutic developments.
Comment 3:
The role of the tumor microenvironment in shaping metabolic and epigenetic changes is absent, despite its growing importance in melanoma biology and therapy resistance.
Response 3:
We thank the reviewer for pointing this out. We have addressed this important aspect by adding a dedicated section titled “Tumor Microenvironment and the Epigenetic–Metabolic Axis in Melanoma” (Section 5). In this new section, we discuss how various components of the tumor microenvironment—including cancer-associated fibroblasts (CAFs), tumor-associated macrophages (TAMs), and secreted metabolites such as lactate—influence both the metabolic state and epigenetic programming of melanoma cells. This section integrates recent findings and further supports the concept of a dynamic and reciprocal interaction between melanoma cells and their surrounding stroma.

Round 2
Reviewer 2 Report
Comments and Suggestions for Authors
The authors have adopted the suggested modifications and hereby improved the quality of their manuscript